# Concomitant Hereditary Spherocytosis and Pyruvate Kinase Deficiency in a Spanish Family with Chronic Hemolytic Anemia: Contribution of Laser Ektacytometry to Clinical Diagnosis

**DOI:** 10.3390/cells11071133

**Published:** 2022-03-28

**Authors:** Joan-Lluis Vives Corrons, Elena Krishnevskaya, Laura Montllor, Valentina Leguizamon, Marta Garcia Bernal

**Affiliations:** 1Red Cell Pathology and Hematopoietic Disorders, Institute for Leukaemia Research Josep Carreras, 08916 Badalona, Spain; ekrishnevskaya@carrerasresearch.org (E.K.); lauramontllor@gmail.com (L.M.); valen.velilla.vl@gmail.com (V.L.); 2Pediatric Hematology Department, University Hospital Mútua Terrassa Terrassa, 08221 Barcelona, Spain; mgarcia@mutuaterrassa.es

**Keywords:** anemia, hemolysis, RBC membrane, pyruvate-kinase deficiency, beta-spectrin, hereditary spherocytosis

## Abstract

Background: Hereditary spherocytosis (HS) and pyruvate kinase deficiency (PKD) are the most common causes of hereditary chronic hemolytic anemia. Here, we describe clinical and genetic characteristics of a Spanish family with concomitant β-spectrin (SPTB) c.647G>A variant and pyruvate kinase (PKLR) c.1706G>A variant. Methods: A family of 11 members was studied. Hematological investigation, hemolysis tests, and specific red cell studies were performed in all family members, according to conventional procedures. An ektacytometric study was performed using the osmoscan module of the Lorca ektacytometer (MaxSis. RR Mechatronics). The presence of the SPTB and PKLR variants was confirmed by t-NGS. Results: The t-NGS genetic characterization of the 11 family members showed the presence of a heterozygous mutation for the β-spectrin (SPTB; c.647G>A) in seven members with HS, three of them co-inherited the PKLR variant c.1706G>A. In the remaining four members, no gene mutation was found. Ektacytometry allowed a clear diagnostic orientation of HS, independently from the PKLR variant. Conclusions: This family study allows concluding that the SPTB mutation, (c.647G>A) previously described as likely pathogenic (LP), should be classified as pathogenic (P), according to the recommendations for pathogenicity of the American College of Medical Genetics and the Association for Molecular Pathology. In addition, after 6 years of clinical follow-up of the patients with HS, it can be inferred that the chronic hemolytic anemia may be attributable to the SPTB mutation only, without influence of the concomitant PKLR. Moreover, only the family members with the SPTB mutation exhibited an ektacytometric profile characteristic of HS.

## 1. Introduction

Hereditary spherocytosis (HS) is the most common cause of hereditary hemolytic anemia (HHA), affecting approximately 1 in 1000 to 2000 individuals of European ancestry [1], and with a relatively high incidence in Spain [2]. It is transmitted as an autosomal dominant character in 75% of cases, and it has become clear that it is a highly heterogeneous disorder, in terms of clinical expression and inheritance [3]. In clinical practice, HS diagnosis is based on a workup which includes clinical, laboratory investigations, family studies, and the exclusion of other causes of hemolytic anemia [4,5]. Laboratory investigations require, at least, three essential conditions: (a) the presence of circulating spherical red cells (spherocytes), (b) a positive eosin-maleimide binding test (EM-binding test), and (c) a characteristic osmotic gradient ektacytometry (OGE) profile. In HS the primary molecular abnormality may affect one or more red cell membrane proteins that, in the order of frequency, as found in Spain, are β-spectrin, Band 3, Ankyrin, α-spectrin, and Protein 4.2 [5]. These proteins are encoded by *SPTB, SLC4A1 ANK, SPTA1*, and *EPB42* genes, respectively. Basically, the inheritance pattern of HS is autosomal dominant and its clinical phenotype is mild to moderate or severe hemolytic anemia. The association of HS with other RBC congenital defects is rare, and it has been described with sickle-cell anemia [6], β-thalassemia [7], glucose-6-phosphate dehydrogenase [8], and pyruvate kinase deficiency [9,10,11,12,13].

Congenital pyruvate kinase deficiency (PKD: EC. 2.7.1.40) is the most common enzyme abnormality in the erythrocyte glycolytic pathway, with an estimated prevalence of 1:20,000 in the global Caucasian population [14,15]. Clinical manifestations of PK deficiency are, in general, only present in subjects homozygous or double-heterozygous for two different allele mutations; whereas, in heterozygous carriers, the enzyme deficiency is usually asymptomatic. In some cases, however, the patients exhibit overt neonatal jaundice or hemolytic crises, triggered by different stress situations such as pregnancy, infections, or concomitant metabolic diseases [16,17]. In Spain, PK deficiency is a very rare disease, and practically all descriptions refer to sporadic cases [18,19,20].

We describe, here, a Spanish family of 11 members, where seven members presented with a HS due to the c.647G>A mutation in the β-spectrin gene (*SPTB*). In three out of these seven members, a concomitant *PKLR* c.1706G>A mutation was found. The *SPTB* variant has been previously described by van Vuren et al. [21] and classified as LP-4, according to the recommendations for pathogenicity of the American College of Medical Genetics and the Association for Molecular Pathology [22]. Here, after the proband’s familial segregation study, we confirm that, on the basis of its phenotype, this variant can be re-classified as P-5. It should be mentioned that genetic analysis is not yet recommended for HS diagnosis, since hundreds of mutations exist; however, the finding of this, same SPTB mutation in a non-related patient of our cohort of patients with chronic hemolytic anemia due to HS [5] provides further support to its P-5 pathogenicity. Finally, since, independently from the coexistence of the heterozygous PK mutation, all the family members with HS have been followed for 6 years since the diagnosis and have exhibited similar clinical phenotype and OGE pattern, it can be assumed that the *PKLR* deficient variant does not modulate the clinical expression of HS, and that ektacytometry is as a valuable clinical tool for the diagnosis of HS.

## 2. Materials and Methods

Blood samples were collected in EDTA-K3 anticoagulant after obtaining informed consent and approval from the Institutional Ethical Committee. All the diagnostic procedures and investigations were performed in accordance with the Helsinki Declaration of 1975. Studies were carried out via a stepwise process, including basic and hemolysis laboratory tests according to Vives Corrons and Aguilar Bascompte [23], RBC enzyme activities were measured by Beutler’s procedure [24], and eosin-5-maleimide (EMA) binding test, as described by King et al. [25] and revisited by King and Zanella [26]. RBC deformability and other rheological parameters were analyzed by osmotic gradient ektacytometry (OGE), using the osmoscan module of the Laser-assisted Optical Rotational Deformability Cell Analyser LoRRca (MaxSis. RR Mechatronics), as previously described [4,5] The parameters used for the OGE evaluation were (1) EI_max_ (maximum elongation index) or the maximal value of RBC deformability; (2) O_min_ (osmotic value where EI is minimal), which represents the conventional RBC osmotic fragility (OF); (3) O_hyper_ (osmotic value corresponding to 50% of the EI_max_), which reflects the cellular hydration status; and (4) AUC (Area under the curve), which reflects RBC deformability in the whole gradient osmolality profile. Genetic identification of SPTB and PKLR mutations was performed by SGene capture, followed by t-NGS, as previously described [4,5].

## 3. Results

The family tree is represented in Figure 1. The proband (ID 1) is a 4 year-old boy, who was admitted for moderate anemia (Hb: 104 g/L), high MCHC (346 g/L), and marked reticulocytosis (237 × 10^9^/L). Few spherocytes were observed in peripheral blood and physical examination showed a splenomegaly of 3 cm under the left costal margin. The hemolysis was confirmed by an increased serum non-conjugated bilirubin (2.50 mg/dL; normal range 0.5–0.8), lactic dehydrogenase (LDH: 767 U/L; normal range: 230–460), and a low serum haptoglobin (<20 mg/dL; normal range: 30–200 mg/dL). The direct antiglobulin test (DAT) was negative, high precision liquid chromatography (HPLC) disclosed hemoglobin abnormalities, and the RBC enzyme activities were found to be within the normal range. There was presence of spherocytosis, together with an increased RBC osmotic fragility (ROF) on both fresh and incubated blood, the positive acidified glycerol lysis test (AGLT: <300 s; normal value: >1800 s), and the reduced binding of eosin 5′ maleimide (EMA-binding test) to −28.43 (Normal Value < 11). Additionally, a mutation of SPTB c.647G>A (p.Arg216Gln) was found by t-NGS analysis, allowing the confirmation of HS. The OGE profile was characteristic of HS, as observed in all seven family members with the SPTB c.647G>A mutation (Figure 2). The proband’s brother (ID 2) and his mother (ID3) were healthy. Proband’s father (ID 4), a 37 year-old man, had displayed moderate anemia and high reticulocyte count since birth until splenectomy, performed at the age of 18 years, which was followed by a normalization of hemoglobin concentration and the reticulocyte count.

The proband’s father (ID4) had a 41-year-old sister (ID5), with a similar clinical and hematological picture, characterized by slight anemia and reticulocytosis since birth that disappeared after splenectomy performed when she was 22 years old. ROF, on both NaCl fresh and incubated blood, a positive acidified glycerol lysis test, a positive EMA-binding test, and an OGE profile were characteristic of HS (Table 1). High performance liquid chromatography (HPLC) rejected the co-inheritance of an hemoglobinopathy, but the measurement of RBC enzyme activities showed a decrease of PK activity (7.5 IU/g Hb, Normal range: 8.4–15.2) and of the PK/HK ratio (6.3, normal range: 7.2–15.6), suggesting the existence of a heterozygous PK deficiency (PKD). The PK activity, also measured in ID 5 children, was found to be within normal range.

ID5 married two times: the first with 41-year-old healthy man (ID6), and the second with a 54-year-old man (ID11). From the first marriage, she had three sons (ID7, ID8, and ID9) and from the second marriage, she had a daughter (ID10). Two of her sons (ID7 and ID10) exhibited a clinical history of neonatal jaundice, positive EMA-binding tests, and typical HS OGE profiles. ID8 suffered, at 11 months of age, an aplastic crisis with severe anemia, due to a B19 parvovirus infection that recovered after a blood transfusion. ID 9 at 11 years-old did not present a clinical history of neonatal anemia or jaundice and maintained a slightly decreased hemoglobin concentration of about 115 g/L throughout his life.

In order to discard an eventual co-inheritance of PK deficiency with HS in this family, a t-NGS study was undertaken in the propositus (D1), his father (ID4), his aunt (ID5), and in all the aunt’s children (ID7, ID8, ID9, and ID10). The t-NGS study demonstrated the existence of the β-Spectrin (SPTB) gene mutation (c.647G>A; p.Arg216Gln) in seven out of 11 family members and of a PKLR gene mutation (c.1706>a; p.Arg569Gln) in three of the seven members with the SPTB mutation, confirming the co-inheritance of HS and PKD in ID5, ID8, and ID9 (Figure 1). PKLR c.1706>a (p.Arg569Gln) mutation has been previously described by Fermo et al. in a 40 year-old women splenectomized for a severe hemolytic anemia [27]. Clinical, hematologic, biochemical, and molecular data of all family members at the time of diagnosis are summarized in Table 1.

## 4. Discussion

Hereditary spherocytosis (HS) is a highly heterogeneous rare anemia in terms of clinical expression, inheritance, and underlying genetic defects. Its estimated prevalence in Europe is about 1:2000 [28], and its incidence in Spain is relatively high [2,29]. In more than 80% of cases, HS has an autosomal dominant inheritance pattern, and its clinical manifestation is a long life chronic hemolytic anemia with variable intensity, from a severe, transfusion-dependent anemia, to an almost silent phenotype, depending on the membrane protein(s) involved, the type of mutation, and other phenotypic modifiers [30,31]. The hallmark of HS is a loss of red cell membrane and a consequent change of the biconcave shape, to a more spherical and dense cell, called spherocyte, that, due to its markedly decreased deformability, is removed from the circulation by the spleen [32].

Hereditary pyruvate kinase deficiency (PKD: EC. 2.7.1.40) is the most common anaerobic glycolysis defect and is a rare, autosomal recessive disease, affecting about 1:20,000 worldwide Caucasian individuals. PKD essentially differs from HS by a relatively more severe clinical phenotype and the absence of circulating spherocytes. PKD hampers cellular energy supply, because glycolysis is the only source of red blood cell ATP production, and the severity of anemia differs considerably, depending on the type of mutation and phenotypic modifiers. Moreover, in some cases, the peripheral blood smear may show few echinocytes and/or tear-drop cells, but RBC morphology is altered to a far lesser extent than in HS [33]. The clinical manifestations of PKD are present in homozygous or double-heterozygous conditions, whereas the heterozygous condition is usually asymptomatic, with the exception of some rare cases with overt neonatal hemolytic anemia and jaundice or with hemolytic crisis triggered by situations of stress, such as pregnancy, infections, or concomitant metabolic diseases [16].

Co-inheritance of HS and PKD is very rare, and, to date, only eight cases have been reported [6,7,8,9,10,11,12,13]. The first case was published in 1970 by Brook and Tanaka [9], and after our first report of this association in Spain [10], we concluded that heterozygous PKD did not modulate the HS clinical expression. Vercellati et al. [11] arrived at the same conclusion after their study of a patient with co-inheritance of HS and a PKLR deficient variant. More recently, however, van Zwieten et al. [12] considered that in a family with HS, the co-inheritance of a partial PKD may aggravate the phenotypic expression of band 3 deficiency, due to the deleterious effect of the ATP decrease. Finally, Andres et al. [13] described a novel association of HS and partial PKD, apparently caused by the loss of membrane-bound PK as the consequence of the impaired structural integrity of the RBC membrane, due to band 3 deficiency.

In this paper, we describe a Spanish family of 11 members, where seven exhibited a dominant HS associated with a moderate chronic hemolytic anemia. The β-Spectrin (SPTB) c.647G>A mutation found in this family has been previously described by van Vuren et al. in a cohort of 95 patients with HS [21] and by our group in another study involving a cohort of 45 patients with several hereditary membranopathies (5).

As shown in the family tree (Figure 1) the proband (ID1), his father (ID4), and his aunt (ID5) showed a similar clinical and hematological picture, characterized by a chronic hemolytic anemia with increased reticulocyte count and increased MCHC. The proband’s mother (ID3) and his brother (ID2) were clinically and hematologically normal. ID4 and ID5 were splenectomized at 18 and 22 years, respectively, and this was followed by a fully compensated hemolysis, without anemia. ID5 married two times with healthy men (ID6 and ID11) and from the first marriage she had three sons (ID7, ID8, and ID9) with chronic hemolytic anemia, more severe in ID7 than in ID8 and ID9 until splenectomy was performed at the age of 12 years (Figure 1). From the second marriage she had a four-month-old daughter (ID 10) with neonatal jaundice, a positive EMA binding test and typical HS OGE profile. The t-NGS study performed in the 11 family members demonstrated the existence of the SPTB mutation in seven members (Figure 1, marked in red), and three of them exhibited a heterozygous PKLR gene mutation (c.1706>a; p.Arg569Gln), confirming the co-inheritance of HS and PKD in patients ID5, ID8, and ID9 (Figure 1, marked in blue).

The question here arises whether the presence of the PKLR c.1706G>A mutation may modulate the clinical expression of HS due to the SPTB c.647G>A mutation. In our previous study of a patient with coinheritance of HS and PK ‘Mallorca’ [10], we concluded that the defective PKLR gene did not influence the HS clinical phenotype. We provide here a further support to this assumption, as can be seen in the large number of family members included in our study where the patients with concomitant SPTB and PKLR mutations (ID5, ID8, and ID9) seem to exhibit the same phenotypic expression as the patients with the SPTB mutation alone. (Figure 1). The slight differences in clinical expression of SPTB mutation between the different members of this family may be explained by the known intra-family variability of clinical phenotype in HS [34] or additional factors (even acquired) that may modify clinical phenotype in these patients [35].

## 5. Conclusions

Our findings provide three main messages: (a) The SPTB mutation (c.647G>A), with a moderate clinical phenotype has to be re-classified as P-5; (b) The tight correlation between the SPTB mutations and the OGE profile provides a further demonstration of the clinical utility of RBC deformability measurement in the diagnosis of HS; and (c) Key glycolytic enzymes activity measurement in patients with chronic hemolytic anemia due to an underlying RBC membrane defect may improve our knowledge of the epidemiologic and genetic characteristics of eventual co-inherited RBC defects, as well as their frequency in the European population.

## Figures and Tables

**Figure 1 cells-11-01133-f001:**
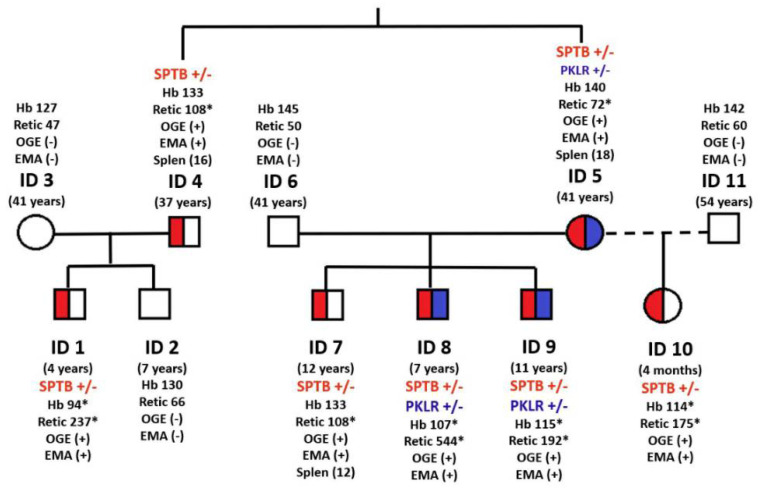
Family tree Scheme 647. G> A (p.Arg216Gln), in beta-spectrin gene; *PKLR* (+/−): heterozygous mutation, c.1706G>A (p.Arg569Gln), in pyruvate kinase gene; Hb: hemoglobin (g/dL); Retics: reticulocytes (×10^9^/L); OGE: osmotic gradient ektacytometry, (+) profile characteristic of HS, (−) normal profile; EMA: EMA-binding test (+) positive; (−) normal; Spleen (age of splenectomy); (*): values outside the reference range.

**Figure 2 cells-11-01133-f002:**
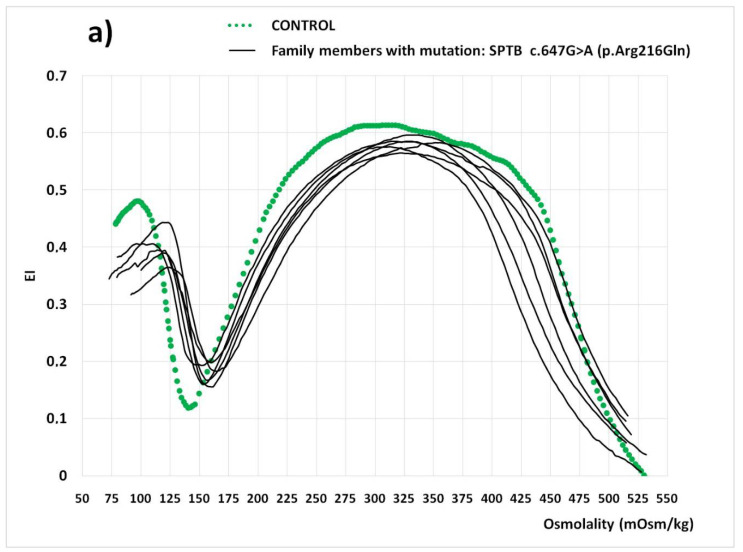
Osmotic Gradient Ektacytometry (OGE) curves. (**a**) Family members with HS (7 out of 11); (**b**) family members without HS (4 out of 11).

**Table 1 cells-11-01133-t001:** Clinical, hematological, and genetic data of the family members.

	ID1	ID2	ID3	ID4	ID5	ID6	ID7	ID8	ID9	ID10	ID11	Reference Values
*Age*	4 years	7 years	41 years	37 years	41 years	41 years	12 years	7 years	11 years	4 months	54 years	
*Neonatal jaundice*	Yes	No	No	No	No	No	Yes	No	No	Yes	No	
*Splenomegaly*	Yes	No	No	Yes	No	No	Yes	Yes	No	No	No	
*Transfusion (No. of units)*	No	No	No	No	No	No	2 (6 and 8 years)	1 (1 year)	No	No	No	
*Hemoglobin (g/dL) **	94	130	127	156	140	145	133	108	115	114	142	121–167
*MCV (fL) **	76.1	81.6	89.9	85.8	86.9	93.3	83.5	87.1	81.5	83.3	93.4	80–94
*MCHC (g/L) **	351	345	323	366	333	343	335	348	368	347	330	310–350
*Reticulocytes (10^9^/L) **	236.7	65.8	46.8	144.1	72.4	50	108.4	544	191.6	175.2	59.7	24–84
*Unconjugated bilirubin (mg/dL)*	2.6	0.1	0.2	0.61	0.60	0.56	2.04	0.98	0.6	5.94	0.54	0.2–0.8
*Haptoglobin (mg/dL)*	<20	65	72	40	78	42	20	20	42	<20	52	30–200
*Lactate Dehydrogenase (IU/L)*	767	250	380	520	266	318	280	582	310	393	183	230–460
*Direct antiglobulin test*	(−)	(−)	(−)	(−)	(−)	(−)	(−)	(−)	(−)	(−)	(−)	(−)
*Splenectomy (years)*	No	No	No	Yes (16)	Yes (18)	No	Yes (12)	No	No	No	No	No
*RBC OSMOTIC FRAGILITYY AND EMA--BINDING TEST *												
*OF in NaCl on fresh blood*	Increased	Decreased	Normal	Normal	normal	normal	Increased	increased	increased	increased	normal	normal
*OF in NaCl on incubated blood*	Decreased	Decreased	Normal	Decreased	increased	normal	increased	increased	increased	increased	normal	normal
*Acidified Glycerol Lysis (s)*	1800	1800	1800	1800	<1800	>1800	>120	<120	<120	>1800	<120	>1800
*EMA-binding test*	–28.43	–12.15	–8.4	–26.2	–15.62	–4.1	–26.23	–18.4	–27.45	–24.09	–2.2	>–11
*OSMOTIC GRADIENT EKTACYTOMETRY (OGE)*												
*EImax*	0.168	0.147	0.151	0.164	0.585	0.619	0.565	0.579	0.586	0.583	0.619	0.60–0.63
*Omin*	158	131	139	157	154	150	162	177	162	164	143	127–159
*Ohiper*	446	439	473	425	461	455	462	476	436	470	468	447–480
*AUC*	145.8	165.4	177.5	134.6	155.8	162.2	146.6	144.7	137.3	147.9	172.1	159–175
*ACTIVITY PYRUVATE KINASE (PK)*												
*PK (IU/gHb)*	11.5	12.31	14.9	8.59	7.95	9.17	9.8	9.61	9.61	14.6	8.73	8.4–15.2
*GENETIC DIAGNOSIS BY t-NGS, >500x*												
*PKLR c.1706G>A (p.Arg569Gln)*				heterozygous	heterozygous			heterozygous	heterozygous			
*SPTB c.647G>A (p.Arg216Gln)*	heterozygous			heterozygous	heterozygous		heterozygous	heterozygous	heterozygous	heterozygous		

*: After splenectomy; MCV: mean cellular volume; MCHC: mean cellular hemoglobin concentration; OF: osmotic fragility; EImax, Omin, Ohyper, AUG: osmotic gradient ektacytometry parameters (see text description).

## Data Availability

Data sharing policies concern the minimal dataset that supports the central findings of a published study. In accordance with journal guidelines, MDPI Research Data Policies generated data should be publicly available and cited in accordance with journal guidelines.

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
