# Peer review of "Concomitant Hereditary Spherocytosis and Pyruvate Kinase Deficiency in a Spanish Family with Chronic Hemolytic Anemia: Contribution of Laser Ektacytometry to Clinical Diagnosis"

_cells, 2022, doi:10.3390/cells11071133_

Round 1
Reviewer 1 Report
The Authors describe a family with co-inheritance of SPTB and PKLR mutations demonstrated by t-NGS, and typical osmoscan patterns. They proposed that the likely pathogenic variant of SPTB should be reclassified as pathogenic and that the co-mutated PKLR does not influence patient phenotype.
The article is interesting and well written. The main concern is the assumption that comutations in other red-cell genes, i.e. PKLR, do not modify clinical phenotype in patients with HS only basing on osmoscan results. In the discussion this may be better commented and this assumption may be smoothened. In fact, studies with different/controversial results are presented, and the Authors may want to comment that several factors (even acquired along life time) may contribute to the clinical phenotype.
Other comments
Introduction:
- it should be mentioned that genetic analysis is not yet recommended for HS diagnosis, since hundreds of mutations exist
- LP-4 and P5 are not explained
- "Finally, since independently....": this assumtpion cannot be completely accepted: in fact the Authors can only state that the concomitant PKLR mutation does not modify osmoscan pattern, but they cannot infer on the contribution of such mutation to patients' phenotype. In fact, it may be speculated that the concomitant mutation might have a clinical contribution in the number and degree of hemolytic crises, iron overload, gallstones formation, etc. that have not been addressed in this study. This should be corrected in the abstract and throughout the paper.
Results
- "HPLC disclosed Hb abnormalities": which abnormalities? or "no" abnormalities?
- "HS was finally confirmed by tNGS...": this is not clinical practice. The Authors may change it into "Additionally, a mutation of ... was found by t-NGS analysis"
- ID9 Hb levels are not normal but slightly decreased, particularly for children
Figure 2b: osmoscan curves of unrelated subjects are also dysplayed? If yes, those of ID6, ID11 and ID3 should be removed
Discussion
- lines 153 to 176 should be removed or summerized since they repeat concepts already stated in the introduction.
- I would suggest to smoothen the concept of "no contribution of concomitant mutations to HS phenotypes"
- I would include a comment on additional factors (even acquired) that may modify clinical phenotype in these patients, consider citing PMID 34627331 "Confounding factors in the diagnosis and management of CHAS.."
Reviewer 2 Report
This is a straightforward, well-written clinical description of a kindred with heterozygosity for a known mutation associated in many cases with hereditary spherocytosis and concurrence presence of a pyruvate kinase mutation. The authors demonstrate fairly conclusively that the clinical phenotype is essentially that of hereditary spherocytosis.
Points of significance are that most likely this particular mutation should be classified as Pathogenic rather than Likely Pathogenic, and that detailed cellular analysis can permit the recognition of the dominant clinical disorder producing a phenotype in patients who are heterozygous for related disorders.
The manuscript is well written. My only comment is that the standard-setting group referred to should be properly capitalized as American College of Medica; Genetics
Author Response
The mention to the American College of Medical Genetics and the Association for Molecular Pathology has been amended and completed
- Zarza R, Moscardó M, Alvarez R, García J, Morey M, Pujades A, Vives-Corrons JL. Co-existence of hereditary spherocytosis and a new red cell pyruvate kinase variant: PK mallorca. Haematologica. 2000 Mar;85(3):227-32. PMID: 10702808
2.Vives-Corrons JL, Krishnevskaya E, Rodriguez IH, Ancochea A. Characterization of hereditary red blood cell 273 membranopathies using combined targeted next-generation sequencing and osmotic gradient ektacytometry. Int J Hematol. 274 2021